# Therapeutic Drug Monitoring for Sirolimus in Children with Vascular Anomalies: What Can We Learn from a Retrospective Study

**DOI:** 10.3390/ph17101255

**Published:** 2024-09-24

**Authors:** Ya-Hui Hu, Yue-Tao Zhao, Hong-Li Guo, Yue Li, Yuan-Yuan Zhang, Jie Wang, Xuan-Sheng Ding, Ji-Jun Zou, Feng Chen

**Affiliations:** 1Pharmaceutical Sciences Research Center, Department of Pharmacy, Children’s Hospital of Nanjing Medical University, Nanjing 210008, China; 2School of Basic Medicine and Clinical Pharmacy, China Pharmaceutical University, Nanjing 210009, China; 3Department of Burns and Plastic Surgery, Children’s Hospital of Nanjing Medical University, Nanjing 210008, China

**Keywords:** sirolimus, vascular anomalies, children, therapeutic drug monitoring, *C*_trough_/Dose ratio

## Abstract

**Objectives**: Sirolimus (SRL), a mammalian target of rapamycin inhibitor, has been widely used to treat patients with vascular anomalies (VAs). The objectives of this study were to summarize the routine blood SRL monitoring data for VAs children, to investigate the factors contributing to the variable blood SRL concentrations and to evaluate the efficacy and safety of SRL therapy. **Methods**: VAs patients with routine blood SRL monitoring from July 2017 to April 2022 at the Department of Burns and Plastic Surgery, Children’s Hospital of Nanjing Medical University were retrospectively collected. Clinical data were obtained from the hospital information system. **Results**: In total, 67 children (35 females) were enrolled. The therapeutic drug monitoring data showed that the range of measured blood trough concentrations (*C*_trough_) was 3.6–46.8 ng/mL. At the initial measurements, only 33% of patients were in the target concentration range (10–15 ng/mL). But this proportion became 54% at the last measurements. The whole blood-*C*_trough_-to-daily dose (*C*_trough_/Dose) ratio was significantly associated with age and body weight (BW). The patients’ laboratory results did not change significantly after SRL treatment. Although the incidence of adverse events was relatively high (44.8%), most of them were mild and tolerable. 70.3% patients had total responses to SRL, whereas 29.7% children exhibited stable disease or progressive disease. No significant differences were found in *C*_trough_ between the total response group and non-response group. **Conclusions**: This retrospective study revealed a high variability in SRL blood concentrations in Chinese children with VAs. Of note, pediatric patients with older age and a higher BW had a lower *C*_trough_/Dose ratio. It is a concern whether the range of 10–15 ng/mL is feasible for Chinese children based only on our study. Further studies recruiting more patients are required to redefine the target reference range for children with VAs.

## 1. Introduction

Vascular anomalies (VAs) are a group of vascular developmental disorders that are classified into vascular tumors and malformations [1]. The International Society for the Study of Vascular Anomalies (ISSVA) approved an updated classification system in May 2018 [2]. Generally, the vascular tumors develop from proliferative changes of vascular endothelial cells, but the vascular malformations are characterized by the abnormal dilation of vessels without proliferation [3]. Due to growth or expansion of VAs, individuals might experience various clinical problems, like chronic pain, recurrent infections, disfigurement, coagulopathies (thrombotic and hemorrhagic), organ dysfunction, and even death [4]. Unfortunately, available therapy options are very limited. Indeed, treatments have been mostly interventional and surgical for the palliation of symptoms. Ideal therapies for patients with VAs would target the key cellular pathways in abnormal vascular proliferation and growth.

The phosphatidylinositol 3-kinase (PI3K)/ protein kinase B (AKT) signaling pathway is critical to cell growth and survival. Of note, this pathway has been demonstrated to govern normal vascular development and angiogenesis [5]. The mammalian target of rapamycin (mTOR), a serine/threonine kinase regulated by the PI3K/AKT, plays an important role in numerous cellular processes, such as cellular catabolism and anabolism, cell motility, angiogenesis, cell growth, and autophagy [6,7,8]. Enhanced mTOR signaling increases expression of the vascular endothelial growth factor, which is a master switch in angiogenesis and lymphangiogenesis [9]. Inappropriate activation of the PI3K/AKT/mTOR pathway has been proven to lead to VA-related tissue overgrowth. Sirolimus (SRL), an mTOR inhibitor, is capable of integrating signals from the PI3K/AKT pathway to coordinate proper cell growth and proliferation [10]. Interestingly, it has emerged as a new medical treatment option for VAs in recent years [11,12].

However, SRL has a narrow therapeutic window, and its disposition exhibits large inter- and intra-patient variability [13]. Therapeutic drug monitoring (TDM) of SRL has been shown to be beneficial for individualizing dose regimens and ensuring efficacy and safety in transplant patients [14,15], but this was not true for patients with VAs. The target whole-blood level for those patients is not well established yet. Indeed, one of the fundamental aims of TDM is the minimization of adverse reactions. Pediatric patients with VAs also need to receive long-term SRL treatment, and safety becomes a major concern. Follow-up requires close monitoring of possible side effects, in addition to infectious complications. Therefore, to balance the need to maintain its efficacy with the avoidance of adverse reactions, frequent blood sampling is required for patients with VAs to ensure that the SRL concentrations in whole blood are within the target therapeutic window (like 5–15 ng/mL).

Of note, the majority of VAs was observed in childhood, including the neonatal period [16]. However, real-world clinical data and TDM implementation for pediatric patients with VAs are very limited. To this end, our study aimed to (1) summarize the routine TDM results, safety, and efficacy of SRL in Chinese children with VAs in our hospital; and (2) identify the potential factors affecting the whole-blood concentrations of SRL.

## 2. Results

### 2.1. Characteristics of Patients

A total of 67 children (35 females) were enrolled in this study (Table 1). The median age and BW of these patients were 3.4 years (IQR 6.0) and 15.0 kg (IQR 12.4), respectively. Notably, approximately 76.7% of patients had low body mass index (BMI) before starting SRL treatment, and the median BMI was 16.2 (range 13.0–26.3) kg/m^2^. 31.3% of patients were diagnosed with vascular tumors, and the others were diagnosed with vascular malformations (68.7%), according to the ISSVA classification for VAs. The most common types of VAs were lymphatic malformations (LMs); approximately 37.3% of patients had an LM, located mostly on the head and neck (13 (56.5%)). In order of decreasing frequency, kaposiform hemangioendothelioma (KHE) is next (19.4%).

### 2.2. Whole Blood C_trough_ of SRL

Overall, 613 measurements of SRL in the whole blood sample were conducted for all the 67 patients, with *C*_trough_ values ranging from 3.6 to 46.8 ng/mL (Figure 1A). A blood SRL level of 10–15 ng/mL was seen as the target therapeutic range in children with VAs in our laboratory. The ratio of the number of measurements within the target range to the total number of measurements was used to calculate the on-target ratio for each individual. The proportion of the on-target ratio ≥50% was 61.2%. The on-target ratio was 0% in five patients and 100% in the other four patients. Subsequently, we listed the initial and the last concentration values for each patient, which were defined as *C*_initial_ and *C*_last_ (Figure 1B), respectively. The median concentration of the *C*_initial_ was 11.9 ng/mL (range: 3.6–33.2 ng/mL), and that of the *C*_last_ was 12.7 ng/mL (range: 4.5–22.1 ng/mL). For *C*_initial_, only 33% of patients achieved the target concentrations. But the percentage became 54% for the *C*_last_.

The distribution of blood concentrations in children with different types of VAs is shown in Figure 2A. Of note, patients with vascular tumors showed higher *C*_trough_/Dose ratio than those of children with vascular malformations except for LMs, both for the *C*_initial_ and the *C*_last_ (*p* = 0.0082 and *p* = 0.0173, respectively; Figure 2B,C).

### 2.3. Age, BW, Sex, BMI, and the C_trough_-to-Daily Dose (C_trough_/Dose) Ratio of SRL

For those *C*_initial_, there was a negative correlation between age and *C*_initial_/Dose ratio (r = −0.7647, *p* < 0.0001; Figure 3A). The *C*_initial_/Dose ratio was significantly higher by 94% (*p* < 0.0001) in children with ≤ 6 years of age (n =45) than children older than 6 years (n = 16). This was also true between BW and *C*_initial_/Dose ratio (r = −0.8144, *p* < 0.0001; Figure 3B). Children with a BW of ≤10 kg (n = 15) showed a higher *C*_initial_/Dose ratio by 3.6-fold and 6.6-fold, compared to those of patients with a BW between 10 to ≤20 kg (n = 26, *p* < 0.0001) and a BW of >20 kg (n = 16, *p* < 0.0001), respectively. And the *C*_initial_/Dose ratio in patients with a BW of >20 kg was 54% of that of children with BWs between 10 to ≤20 kg. Nevertheless, males and females were exposed to similar SRL levels (Figure 3C). No correlation was found between BMI and *C*_initial_/Dose ratio of SRL (r = 0.1191, *p* = 0.3607).

For *C*_last_, impressively, the significant negative correlation between age and *C*_last_/Dose ratio was still retained (r = −0.7245, *p* < 0.0001; Figure 3D). The same was observed for BW and *C*_last_/Dose ratio (r = −0.7016, *p* < 0.0001; Figure 3E). However, no significant difference was found in *C*_last_/Dose ratio between females and males (Figure 3F). Only a negligible correlation was observed between BMI and *C*_last_/Dose ratio (r = 0.1087, *p* = 0.4671).

Additionally, we also performed various parallel analyses after stratification according to disease type (Table 2). Notably, the *C*_trough_/Dose ratios in children with vascular tumors were associated strongly with age and BW.

### 2.4. Concomitant Medications and the C_trough_/Dose Ratio of SRL

To answer whether co-administration with other medications contributes to the *C*_trough_/Dose ratio, we evaluated the influences of various concomitant medications on blood SRL levels. No reported cytochrome P450 enzyme (CYP450) inhibitors or inducers were found in the combination of drugs used in the children. The most frequently used drugs were ipratropium bromide, terbutaline sulfate, and budesonide. Overall, the effects of concomitant drugs on *C*_trough_/Dose ratio were negligible.

### 2.5. Laboratory Test Results

We compared the last laboratory test results to the initial laboratory test results for each individual (Figure 4). The laboratory biomarkers did not change significantly after SRL treatment. Of note, the ratio of PLT was particularly high in several children because these patients had low values of PLT at the initial examinations due to disease (e.g., Kasabach–Merritt phenomenon) or neonatal thrombocytopenia, but was not associated with SRL therapy.

It is worth noting that the reference intervals for blood cell analysis and biochemical tests in children vary with age. Thus, we stratified the results of laboratory tests by age groups. The initial and last test results were compared within each age group. The results showed that the changes in six of all nineteen biomarkers were statistically different (Figure 5). MCH, MCHC, AST, and Cys-C decreased after SRL medication (*p* = 0.0044, Figure 5B; *p* = 0.0200, Figure 5C; *p* = 0.0449, Figure 5D; *p* < 0.0001, Figure 5F; respectively). In contrast, HGB and TC increased (*p* = 0.0391, Figure 5A; *p* = 0.0105, Figure 5E). There were no significant differences in other results.

### 2.6. The Incidence and Severity of Adverse Events

Adverse events of SRL were reported in 30 patients (44.8%), including hyperlipidemia, gastrointestinal reactions, upper respiratory tract infections, mucositis, and increases in liver enzymes. Each adverse event was evaluated according to the Common Terminology Criteria for Adverse Events (CTCAE), version 5.0.

Hyperlipidemia had the highest incidence (28.4%), documented in 19 patients, and mainly manifested in the increase in triglycerides and cholesterol. Among them, 18 patients developed hypertriglyceridemia (fourteen patients with grade 1 and four patients with grade 2) and one patient developed grade 1 hypercholesterolemia. A grade 3 transitory elevation of liver enzyme activity was observed in one patient. The incidence of gastrointestinal reactions and upper respiratory tract infections were 16.4% and 14.9%, respectively. Primarily adverse events reported like diarrhea, vomiting, and mucositis were classified as grade 1 or 2. All patients could tolerate the SRL medications. Overall, even if adverse events were common, most of them were mild to moderate and resolved spontaneously.

### 2.7. Clinical Outcome

On the SRL effect analysis, 54 patients were eligible. The outcomes were recorded in Table 3. In the two subcategories of VAs, the median duration of SRL treatment for patient with vascular tumors was 10.8 months (IQR 11.8), and 16.9 months (IQR 17.4) for patients with vascular malformations. After SRL treatment, 12 (22.2%) and 26 (48.1%) patients achieved GR and PR, respectively, and the overall response rate was 70.3%. Nevertheless, none of the patients had exhibited a CR during the observational period. No significant differences were found in the TR rate between the vascular tumors group and vascular malformations group (70.0% vs. 70.6%; *p* = 0.964), nor between the KHE group and the LM group (61.5% vs. 65.2%; *p* > 0.999). Interestingly, there were no significant differences between TR and non-response groups (SD and PD) regarding *C*_last_ (*p* = 0.959).

## 3. Discussion

Since both the efficacy and the side effects of SRL have been associated with its whole blood concentration, guidelines for its monitoring and recommendations for its target concentrations in the context of organ transplantation have been proposed [17,18,19]. However, this approach might not reflect the true response of individual patients with VAs because the wide pharmacokinetic interindividual variability means that similar concentrations of the same drug can produce different effects in each patient [20]. In the clinic, indeed, SRL was first tested on an off-label basis in patients with complex life-threatening Vas, and thereafter, clinical drug trials targeting VAs have increased during the past 10 years [21,22]. Up to now, the routine TDM data of SRL for pediatric patients with VAs has been limited, especially for Chinese children. This retrospective study investigated the current situation of TDM of SRL in children with VAs in our hospital. The factors influencing SRL *C*_trough_/Dose values were also evaluated. Particularly, we reported, for the first time, the TDM reference range of SRL for those children.

The first major finding of this retrospective study was that less than 50% concentration measurements of SRL were within the target therapeutic range (i.e., 10–15 ng/mL), even though those patients undergoing long-term TDM (Figure 1).

For patients with VAs, a *C*_trough_ range from 5 to 15 ng/mL was usually considered sufficient, but this was based on the experiences from kidney transplants [11]. A Phase II trial kept *C*_trough_ of SRL at 10–15 ng/mL and reached the overall efficacy of SRL in the treatment of VAs [4]. A systematic review finally enrolled 73 studies focusing on SRL therapy for VAs and revealed that SRL improved the prognosis of VAs [23]. Of note, the goal range of blood SRL level was most frequently 10–15 ng/mL (38.3%) and 5–15 ng/mL (38.3%) in pediatric patients. There are very limited data available for Chinese children. Therefore, it is a practical problem whether the blood SRL concentration maintained at 10–15 ng/mL or even at 5–15 ng/mL is feasible for pediatric patients with VAs in China.

With whole blood SRL level of 10–15 ng/mL as the target therapeutic range, our data confirmed that SRL can effectively reduce the size of lesions in patients with VAs. Of the 54 patients who responded to treatment (Table 3), improvements were observed in 70.3% of patients. After SRL therapy, GR and PR rates of 22.2% and 48.1%, respectively, whereas 14.8% had SD and 14.8% had PD. However, no one achieve CR in the current study. Notably, there were no significant differences in *C*_trough_ between the total response group (GR + PR) and non-response group (SD + PD). Considering the combination of clinical efficacy, side effects, and SRL concentration, such systemic exposure levels may be appropriate for these children. That is, with the help of the TDM approach, an optimal concentration range was achieved for them. We need to collect more data to redefine the target reference of TDM that is suitable for Chinese children. Collectively, the necessity to re-evaluate the target therapeutic reference range also remains open for discussion.

A marked variability in blood SRL levels in pediatric patients with VAs was observed. There was also a large fluctuation in the values of each monitoring for the same patient. Of note, a lower *C*_trough_/Dose ratio was associated with older age and a higher BW (Figure 3). Indeed, various factors like age, BW, and drug–drug interactions contributed to the inter- and intra-individual variability in whole-blood concentrations of SRL.

In vitro hepatic ontogeny functions showed an increase in CYP3A4 activity and/or abundance over age [24]. A consistent observation was found in clinical studies of drugs metabolized in the liver, which was described as an age-dependent increase in plasma clearance in children younger than 10 years of age [25]. Hence, a decreased *C*_trough_/Dose with age was in line with the ontogeny of CYP enzymes involved in SRL metabolism, evidenced by a population pharmacokinetics (PPK) study [26]. Of note, Mizuno et al. [27] found the trajectory of simulated SRL clearances increased with age and identified age-appropriate SRL dosing regimens for neonates and infants, although a fixed initial dose of SRL was commonly used in patients with VAs.

The effect of BW on blood SRL levels is still limited. Scott et al. identified a significant positive correlation between total BW and SRL clearance in a PPK study of 44 pediatric patients with neurofibromatosis type 1 [15]. In a recent PPK study, Cheng et al. revealed that BW had a positive influence on CL/F in 27 children with immune cytopenia [28]. Therefore, BW may influence *C*_trough_/Dose by affecting clearance.

In our study, the median BMI was 16.2 (range 13.0–26.3) kg/m^2^ and 16.7 (range 12.2–22.4) kg/m^2^ before and after SRL use, respectively. Some previous studies have reported negative growth impacts associated with SRL in renal transplant patients [29,30]. Intriguingly, a fairly high proportion (76.7%) of patients with low BMI (<18.5 kg/m^2^) were observed before SRL administration, which may be related to the primary disease and prior treatment. In fact, the impact of SRL use on BMI is minimal; no significant differences were found in BMI before and after SRL treatment (*p* = 0.9710). This finding was consistent with Wang et al.’s [31]. Also, the correlation between BMI and *C*_trough_/Dose ratio was also evaluated during SRL treatment. In the present study, BMI had no relevant effects on the SRL *C*_trough_/Dose ratio at either initial or last measurements.

In addition, previous studies have demonstrated drug–drug interactions between SRL and various medications, such as azole antifungals, rifamycin and cyclosporine [32,33,34]. In the present study, the assessment of potential drug–drug interactions between SRL and other drugs was performed. But there were no concomitant medications that would alter the disposition of SRL.

Interestingly, it seemed that the pathological types of VAs contribute to the variable *C*_trough_/Dose ratio of SRL among those patients (Figure 2). Particularly, children with vascular tumors showed higher *C*_trough_/Dose ratio (including *C*_initial_ and the *C*_last_) than cases with vascular malformations, excluding LMs. Indeed, inter- and intra-individual variability appeared to be very large in children with high SRL exposure, whereas measurements in children with low exposure appeared to be relatively concentrated. This marked variability is thought to be the real cause of the differences between the two groups, but it has nothing to do with the pathological type in essence.

Another relevant finding of our study was that SRL had little impact on the physiological functions of those children (Figure 5). In the present study, the laboratory test results of the children were relatively stable. The ratios of the last laboratory test results to those of the initial laboratory tests were generally less than 3 (Figure 4). However, several biomarkers increased or decreased to some extent after SRL. For instance, TC increased by 0.27 mmol/L (Figure 5E) and Cys-C dropped from 1.09 to 0.92 mg/L (Figure 5F), respectively.

In our study, more than half of the children had blood SRL concentrations that were out of the target range. That might be the reason for the higher incidence of adverse events (44.8%) compared with the previous studies [23,35]. The most common adverse events of SRL include oral mucositis, upper respiratory tract infection, increases in liver enzymes, and dyslipidemia [23,36,37,38]. Despite this, regular monitoring of SRL is generally considered safe [39]. In the current study, hyperlipidemia was the most common adverse event (28.4%). Moreover, oral mucositis, gastrointestinal reactions and upper respiratory tract infections were also greater than 10%. Although the incidence of adverse events was relatively high, most of them were mild and tolerable.

One more question needs to be further discussed. The age-specific reference intervals (‘normal ranges’) are commonly used in medical screening and clinical practice [40,41]. Interestingly, no significant changes in most biomarkers were found across different age groups during the SRL treatment periods (Figure 5). Indeed, in the few reports, some studies evaluated whether a biochemical indicator was affected by comparing baseline and post-treatment values [35,38], but they did not stratify the study populations according to age. Therefore, the true impact of age on biomarkers requires further concern in the future.

One of the major strengths of our study was the routine blood SRL monitoring as early as 2018 for childhood VAs in China, so we have the chance to redefine the target reference of SRL and to evaluate the potential impactors on the *C*_trough_/Dose ratio of SRL in our subjects. Indeed, VAs are not a group of high-prevalence diseases but could even be considered as a group of rare disorders. Fortunately, our hospital had the opportunity to treat pediatric patients from all over the country, and our laboratory could monitor the drug concentration of SRL for those patients earlier, which created the conditions for us to achieve the purpose of this research. Such real-world clinical data are very limited so far, but they are useful for improving the implementing of TDM for SRL in this population for pediatricians and clinical pharmacists when they try to tailor SRL dosages for precision therapy.

However, our study has several limitations. Firstly, this was a single-center and retrospective study with a small sample size, although our hospital had admitted the majority of pediatric patients from Nanjing and surrounding cities. Thus, our findings as a reference should be interpreted with caution. Secondly, the evaluation of the efficacy of SRL on VAs was basically based on the changes of VA volumes, which were measured using volumetric MRI analysis, but not all patients are able to provide these data. In addition, the limitation due to randomization and eligibility criteria of our retrospective study should be a concern. Nevertheless, this study provided invaluable clinical experiences for the use of SRL in children with VAs in China, especially for the implementation of TDM for those patients.

## 4. Patients and Methods

### 4.1. Patients

This study retrospectively included pediatric patients (<18 years) who were diagnosed with VAs and received SRL treatment without other concurrent therapeutic regimens at the Department of Burns and Plastic Surgery, Children’s Hospital of Nanjing Medical University from July 2017 to April 2022 (Figure 6). The initial dose of SRL was 0.1 mg/kg/d or 0.8 mg/m^2^/d, twice daily at 12 h interval, then adjusted according to the target therapeutic window of 10–15 ng/mL. Patients were excluded if: (1) they were treated with SRL, but the routine whole blood measurements were <2 times; (2) their dosage information was incomplete or unavailable; (3) they used oral solutions instead of tablets. The study protocol was approved by the Children’s Hospital of Nanjing Medical University Ethics Committee (protocol number 202206114-1). Due to the retrospective nature of this study, written consents were waivered.

### 4.2. Data Collection

Various data on sex, age, height, body weight (BW), types of VAs, concomitant drugs, adverse events, and laboratory test results including routine blood examinations, liver function, blood lipids and kidney function were collected. The diagnoses of VAs classification were based on typical clinical manifestations, physical examination, imaging tests, and pathological examination. Routine blood tests included red blood cell count (RBC), platelet count (PLT), white blood cell count (WBC), hematocrit (HCT), hemoglobin (HGB), mean corpuscular hemoglobin (MCH), and mean corpuscular hemoglobin concentration (MCHC). Liver function biomarkers like alanine aminotransferase (ALT), aspartate aminotransferase (AST), albumin (ALB), total bilirubin (TBIL), and direct bilirubin (DBIL) were measured. Blood lipid levels such as total cholesterol (TC), triglycerides (TG) and high-density lipoprotein cholesterol (HDL-C) were collected. Kidney function biomarkers included blood urea nitrogen (BUN), serum creatinine (Scr), uric acid (UA), and cystatin C (Cys-C). Importantly, the dosage data when routinely monitoring the trough concentrations (*C*_trough_) of SRL were also collected.

### 4.3. Routine Therapeutic Monitoring of SRL

Whole blood samples from pediatric patients with SRL treatment were taken at 3 to 4 days after the load dose or 7 to 14 days after the maintenance dose was adjusted. Whole blood trough concentration of sirolimus was monitored 0.5 h before administration. Appropriately 1 mL peripheral blood samples were collected into ethylenediaminetetraacetic acid (EDTA)-K2 tubes and stored at 2–8 °C for no more than 5 days.

Assay performance can be obtained from our previous published study [17]. Sample clean-up was required prior to measurement on the analyzer. This was accomplished by adding 50 μL of the sample pretreatment reagent (Siemens Healthcare Diagnostics Inc. Newark, NJ, USA) and 200 μL of methanol to 200 μL of real whole blood samples, calibrators, or controls in microcentrifuge tubes. The resultant samples were then vortexed for 5 min, followed by keeping at room temperature for another 2 min, and then centrifuged at 12,000 rpm for 5 min at 4 °C. The resulting supernatant was decanted and measured on an automated enzyme immunoassay analyzer (SIEMENS, Munich, Germany). The time to run each individual sample was approximately 20 min. Using SRL calibrators analyzed in duplicate, the data were fitted to a parametric logit mathematical equation. Concentration results were calculated by the analyzer automatically. The analyzer has a reportable concentration range for SRL concentration of 3.50 to 30.0 ng/mL based on detection limit and instrument sensitivity. For results above 30.0 ng/mL, the samples were re-analyzed after dilution and the output result was multiplied by the dilution factor to obtain the true concentration of the sample.

### 4.4. Outcome Measures and Definitions

All patients evaluated for efficacy received treatment for ≥3 months. The changes in VAs were measured by radiologists using magnetic resonance imaging (MRI) scans or ultrasonography or computed tomography (CT) for analysis and interpretation in the SRL interventional period. If the intricate shapes of the lesions were difficult to measure, or there were no valid VA volume data, the response was evaluated comprehensively by the clinician. Assessment bases included the degree of shrinkage of the lesion, the change in the number and color of the lesions, coagulation function, pain, and so on.

In this study, the response was defined as complete response (CR), good response (GR), partial response (PR), stable disease (SD), or progressive disease (PD) by referring to previous studies [35] and making minor adjustments. CR was defined as complete resolution of measured VAs (100%) during the observational period. GR was defined as a lesion size decrease of ≥75% but <100% compared with the baseline VA volume or combined with ultrasound or CT and laboratory data, with the clinical symptoms remarkably relieved. PR was defined as ≥20% but <75% reduction in lesion size compared with the baseline data based on MRI, ultrasonography, or CT, accompanied by improvements in most clinical symptoms. SD was defined as a lesion size increase of <20% but decrease of <20% in volumes of VA lesions or symptoms, and/or functional disability of patients was not improved after SRL treatment. PD was defined as an increase of ≥20% in lesion size compared with baseline data, according to imageological examination and laboratory test results. Total responses (TRs) included CRs, GRs, and PRs.

### 4.5. Statistical Analysis

Data analyses were all performed using GraphPad Prism (version 9.4, GraphPad Software, La Jolla, CA, USA). Shapiro–Wilk tests were used to assess normality. Non-normally distributed continuous variables were expressed as median with an interquartile range, and means and standard deviations were described for normally distributed continuous variables. Demographic data and clinical characteristics were described as the frequency for categorical variables. Correlations were tested by Spearman’s correlation coefficient analysis. Comparisons between two groups were performed using Mann–Whitney U tests or unpaired t tests. Differences between multiple groups were assessed by Kruskal–Wallis tests and Dunn’s tests. Paired t tests or Wilcoxon signed ranks tests were used to compare the clinical biochemical data within various age groups. Comparisons of clinical outcomes between subgroups (more than four patients) were performed using the Fisher exact test or χ^2^ test. A *p* value of <0.05 was considered statistically significant.

## 5. Conclusions

In summary, this retrospective study found that blood SRL concentrations were highly variable in children with VAs, and only about 50% of the concentration values were within the target therapeutic reference range, even with long-term TDM. We also identified age and BW as contributing factors to *C*_trough_/Dose ratio of SRL in children with VAs. Briefly, children with older age and a higher BW had a lower *C*_trough_/Dose ratio. Based on the data in our hands, no correlation was observed between efficacy and SRL concentration. It is a concern whether the range of 10–15 ng/mL is feasible for Chinese children. In addition, SRL had a minor effect on the physiological functions of pediatric patients, and the treatment could be considered safe. Arguably, better TDM and dose adjustment schedules need to be implemented to optimize sirolimus therapy in children with VAs. Further investigations are needed in the future.

## Figures and Tables

**Figure 1 pharmaceuticals-17-01255-f001:**
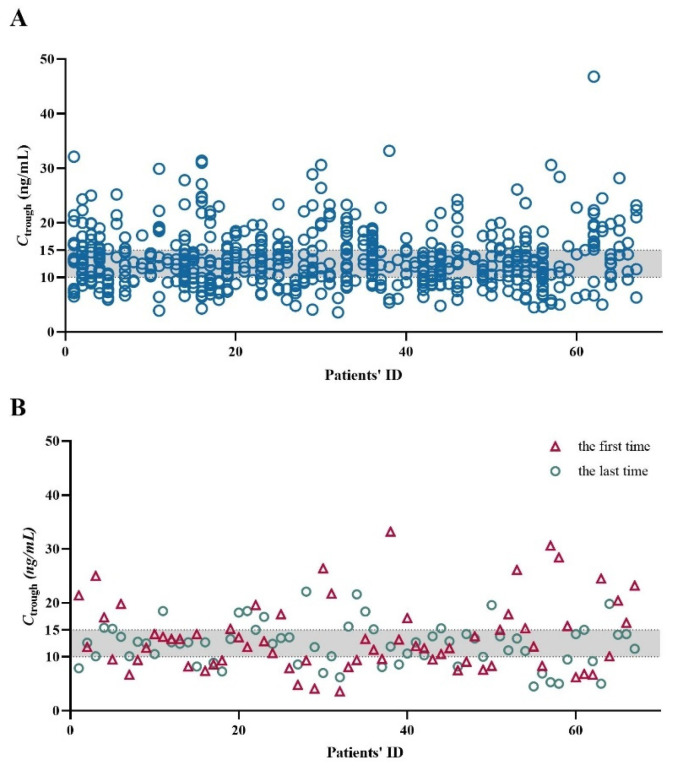
Whole blood SRL trough concentrations (*C*_trough_, ng/mL) in children with Vas. The x-axis shows the number of patients. The grey-filled areas indicate the therapeutic range of SRL (10–15 ng/mL) in our laboratory. (**A**) 613 concentration values in all 67 children. (**B**) The initial measurements for every patient (n = 67).

**Figure 2 pharmaceuticals-17-01255-f002:**
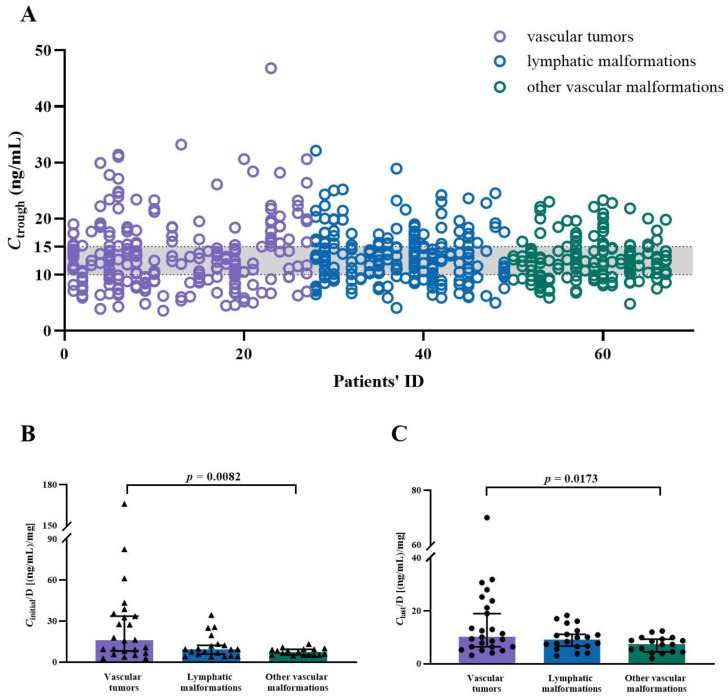
Whole blood SRL trough concentrations (*C*_trough_, ng/mL) in children with different types of VAs and comparisons of *C*_trough_/Dose ratios [(ng/mL)/mg] in different subtypes. (**A**) The distribution of *C*_trough_ values in children with vascular tumors, lymphatic malformations, and other vascular malformations. (**B**) Comparisons of *C*_initial_/Dose ratio in patients with vascular tumors, lymphatic malformations, and other vascular malformations. (**C**) Comparisons of *C*_last_/Dose ratio in patients with vascular tumors, lymphatic malformations, and other vascular malformations.

**Figure 3 pharmaceuticals-17-01255-f003:**
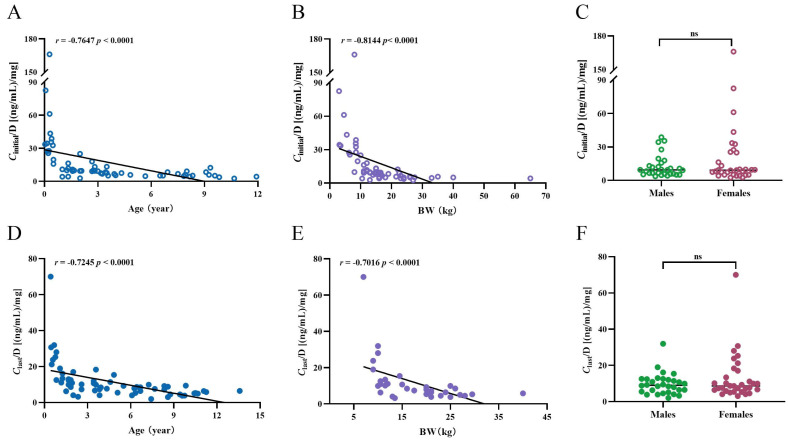
The *C*_trough_/Dose ratios [(ng/mL)/mg] of SRL at the initial and the last measurements in children with VAs. (**A**) Correlation between *C*_trough_/Dose ratio and age at the initial measurements. (**B**) Correlation between *C*_trough_/Dose ratio and BW at the initial measurements. (**C**) Comparison of *C*_trough_/Dose ratio in males and females at the initial measurements. (**D**) Correlation between *C*_trough_/Dose ratio and age at the last measurements. (**E**) Correlation between *C*_trough_/Dose ratio and BW at the last measurements. (**F**) Comparison of *C*_trough_/Dose ratio in males and females at the last measurements.

**Figure 4 pharmaceuticals-17-01255-f004:**
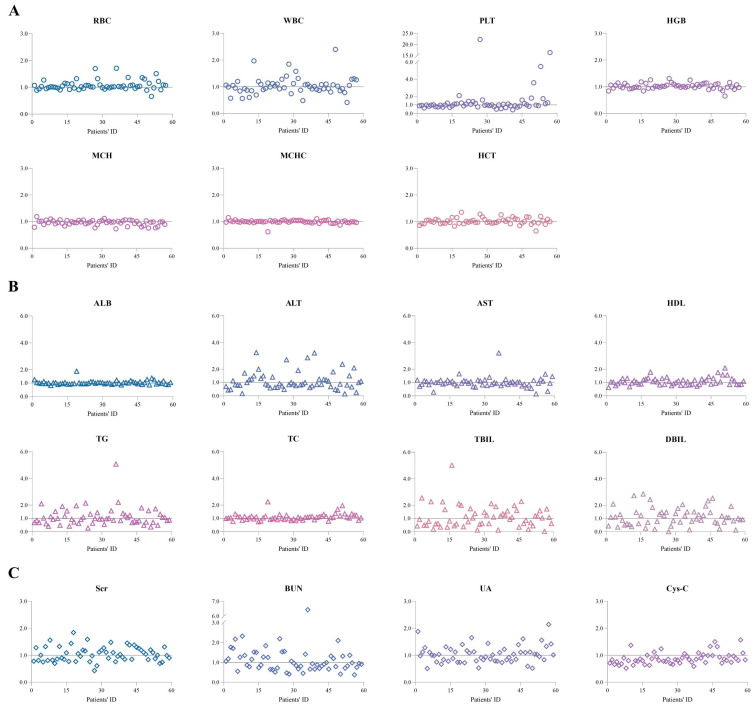
The ratios of the last laboratory test results to the initial laboratory test results for each patient throughout the collected medical records. (**A**) Blood cell analysis. (**B**) Liver function indicators and blood lipids. (**C**) Kidney function indicators. Abbreviations: RBC, red blood cell count; WBC, white blood cell count; PLT, platelet count; HGB, hemoglobin; MCH, mean corpuscular hemoglobin; MCHC, mean corpuscular hemoglobin concentration; HCT, hematocrit; ALB, albumin; ALT, alanine aminotransferase; AST, aspartate aminotransferase; HDL, high-density lipoprotein; TG, triglycerides; TC, total cholesterol; TBIL, total bilirubin; DBIL, direct bilirubin; Scr, serum creatinine; BUN, blood urea nitrogen; UA, uric acid; Cys-C, cystatin C.

**Figure 5 pharmaceuticals-17-01255-f005:**
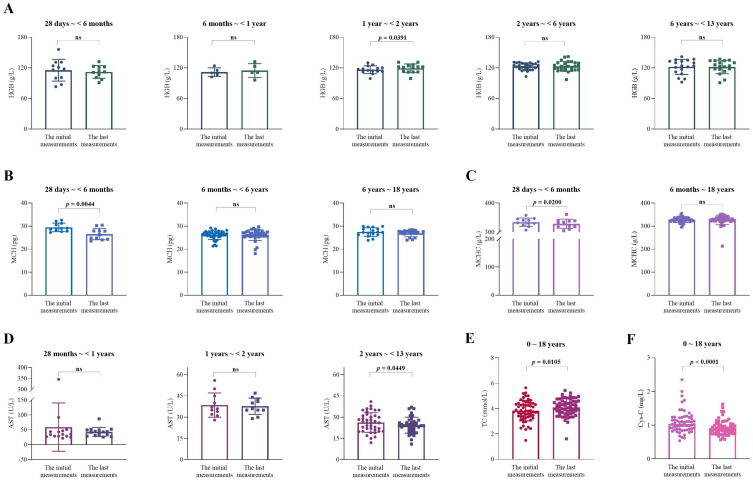
Comparisons of the initial and last test results of patients within each age group. (**A**) Hemoglobin (HGB, g/L). (**B**) Mean corpuscular hemoglobin (MCH, pg). (**C**) Mean corpuscular hemoglobin concentration (MCHC, g/L). (**D**) Aspartate aminotransferase (AST, U/L). (**E**) Total cholesterol (TC, mmol/L). (**F**) Cystatin C (Cys-C, mg/L).

**Figure 6 pharmaceuticals-17-01255-f006:**
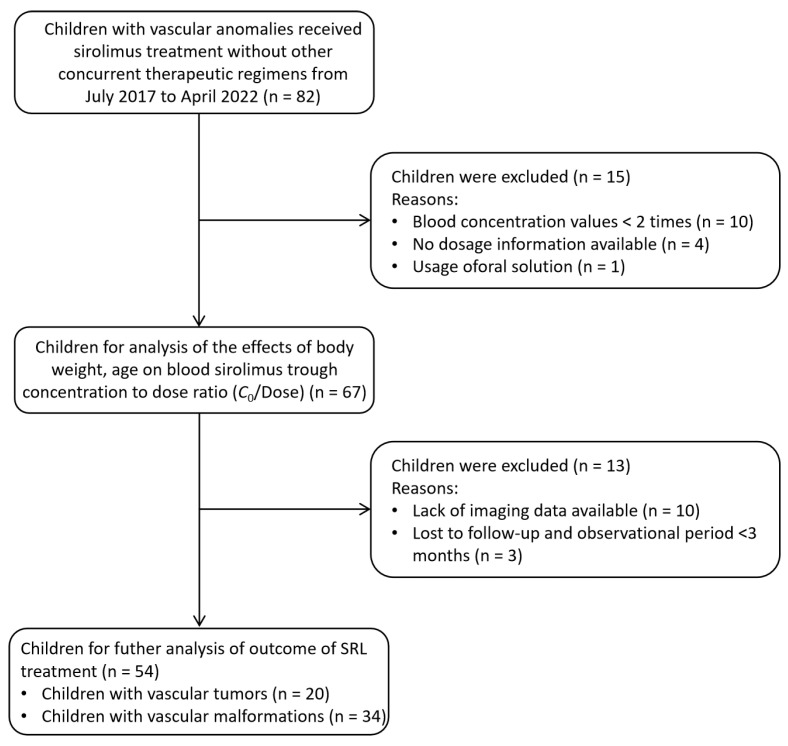
Numbers of patients who were eligible for this study.

**Table 1 pharmaceuticals-17-01255-t001:** Clinical characteristics of patients.

Characteristics	Value
Age (year)	
Median [IQR]	3.4 [6.0]
Range	0.0–13.6
Sex	
Females	35
Males	32
Body weight (kg)	
Median [IQR]	15.0 [12.4]
Range	3.0–65.0
Body mass index (kg/m²)	
Median [IQR]	16.2 [3.4]
Range	13.0–26.3
Type of vascular anomalies, n (%)	
Vascular tumors	21 (31.3%)
Vascular malformations	46 (68.7%)

**Table 2 pharmaceuticals-17-01255-t002:** Age, BW, sex, and *C*_trough_/Dose ratio of SRL in children with different types of vascular anomalies.

Type of VAs	The Initial Measurements			The Last Measurements		
	Age and *C*_trough_/D	BW and *C*_trough_/D	Sex and *C*_trough_/D	Age and *C*_trough_/D	BW and *C*_trough_/D	Sex and *C*_trough_/D
Vascular tumors	Correlation:	Correlation:	No difference	Correlation:	Correlation:	No difference
	r = −0.8741	r = −0.9126		r = −0.8205	r = −0.7634	
	*p* < 0.0001	*p* < 0.0001		*p* < 0.0001	*p* = 0.0014	
	A difference between age ≤ 6 years and age > 6 years (*p* = 0.0006).	A difference between BW ≤ 10 kg and BW between 10 to ≤20 kg (*p* = 0.0031), and a difference between BW ≤ 10 kg and BW ≥ 20 kg (*p* < 0.0001).		A difference between age ≤ 6 years and age > 6 years (*p* = 0.0071).	A difference between BW ≤ 10 kg and BW between 10 to ≤20 kg (*p* = 0.0081), and a difference between BW ≤ 10 kg and BW ≥ 20 kg (*p* < 0.0076).	
Lymphatic malformations (LMs)	Correlation:	Correlation:	No difference	Correlation:	Correlation:	No difference
	r = −0.4802	r = −0.6705		r = −0.5962	r = −0.7311	
	*p* = 0.0321	*p* = 0.0023		*p* = 0.0055	*p* = 0.0252	
	No difference between age ≤ 6 years and age > 6 years.	A difference between BW ≤ 10 kg and BW between 10 to ≤20 kg (*p* = 0.0148), and a difference between BW ≤ 10 kg and BW ≥ 20 kg (*p* = 0.0053).		A difference between age ≤ 6 years and age > 6 years (*p* = 0.0037).	Analysis could not be performed because the majority of patients weighed in 10 ≤ 20 kg.	
Other vascular malformations	Correlation:	Correlation:	No difference	Correlation:	No correlation	No difference
	r = −0.5022	r = −0.6575		r = −0.5515		
	*p* = 0.0399	*p* = 0.0056		*p* = 0.0217		
	No difference between age ≤ 6 years and age > 6 years.	A difference between BW > 20 kg and BW between 10 to ≤20 kg (*p* = 0.0221). This population could just be divided into these two groups.		No difference between age ≤ 6 years and age > 6 years.	Analysis could not be performed because all of the patients weighed > 20 kg.	

Note: BW, body weight; *C*_trough_/D, *C*_trough_/Dose ratio.

**Table 3 pharmaceuticals-17-01255-t003:** The efficacy measure for SRL treatment on 54 patients with vascular anomalies.

Diagnosis		Patients	SRL Duration (Months)	*C*_trough_ (ng/mL)	Outcome			
					GR	PR	SD	PD
vascular tumors	KHE	13 (24.1)	11.2 (6.9–16.0)	12.5 (9.6–17.7)	4 (30.8)	4 (30.8)	3 (23.1)	2 (15.4)
	other	7 (13.0)	10.8 (4.9–21.3)	10.5 (9.1–15.1)	2 (28.6)	4 (57.1)	1 (14.3)	0 (0)
vascular malformations	LMs	23 (42.6)	21.9 (12.3–34.0)	12.7 (10.0–15.6)	4 (17.4)	11 (47.8)	2 (8.7)	6 (26.1)
	BRBNS	1 (2.0)	15.7	13.4 (10.5–16.4)	0 (0)	1 (100)	0 (0)	0 (0)
	CVM	1 (2.0)	10.4	13.0 (11.3–17.4)	1 (100)	0 (0)	0 (0)	0 (0)
	other	9 (16.7)	14.7 (8.7–19.0)	11.8 (9.5–14.6)	1 (11.1)	6 (66.7)	2 (22.2)	0 (0)
Total		54 (100)	13.8 (10.3–22.3)	13.2 (6.9–16.0)	12 (22.2)	26 (48.1)	8 (14.8)	8 (14.8)

Note: Data are median (first to third quartile), n (%), unless stated otherwise. GR, good response; PR, partial response; SD, stable disease; PD, progressive disease; KHE, kaposiform hemangioendothelioma; LMs, lymphatic malformations; BRBNS, blue rubber bleb nevus syndrome; CVM, combined vascular malformations.

## Data Availability

The original contributions presented in the study are included in the article, further inquiries can be directed to the corresponding author.

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
