# Peer review of "Therapeutic Drug Monitoring for Sirolimus in Children with Vascular Anomalies: What Can We Learn from a Retrospective Study"

_pharmaceuticals, 2024, doi:10.3390/ph17101255_

Round 1
Reviewer 1 Report
Comments and Suggestions for Authors
The article submitted by Hu et al. is a retrospective study devoted to the search of factors influencing sirolimus concentration in children with vascular anomalies. The manuscript is written and arranged very well, the data are presented and discussed extensively, so, I have only minor comments on the submission. These are following:
1) Figure 1 is duplicated.
2) Section 3.1 contains characteristics of patients and therefore should be appended with section 2.1.
3) Regarding Fig. 2, I suggest following: 1) remove Fig. 2B and 2D; 2) in Fig. 2C, make all red triangles uniform.
4) Please add the legend to Figs. 3B and 3D indicating initial and last measurements.
5) I recommend adding the explanation of the measured parameters (RBC, WBC etc.) to the Figure caption.
6) Page 14, line 345: "Hence, a decreased C0/Dose with 344 age was in line with the ontogeny of CYP enzymes involved in SRL metabolism. And a 345 population pharmacokinetics..." This sentence seems to be accidentally divided into two.
As the comments are more technical, I recommend minor revision.
Author Response
Reviewer 1: The article submitted by Hu et al. is a retrospective study devoted to the search of factors influencing sirolimus concentration in children with vascular anomalies. The manuscript is written and arranged very well, the data are presented and discussed extensively, so, I have only minor comments on the submission. These are following:
- Figure 1 is duplicated.
Exactly. The duplicated Figure 1 has been removed.
- Section 3.1 contains characteristics of patients and therefore should be appended with section 2.1.
Section 2.1 belongs to “Patients and Methods”, and Section 3.1 describes the characteristics of patients in the “Results” section.
- Regarding Fig. 2, I suggest following: 1) remove Fig. 2B and 2D; 2) in Fig. 2C, make all red triangles uniform.
Yes. The Figure 2 has been revised.
- Please add the legend to 3B and 3Dindicating initial and last measurements.
Yes. We have revised the ordinate titles of the Figs. 3B and 3C.
- I recommend adding the explanation of the measured parameters(RBC, WBC etc.) to the Figure caption.
We added the explanation of the measured parameters in Figure 5 caption as follows:
Figure 5. The ratios of the last laboratory test results to the initial laboratory test results for each patient throughout the collected medical records. (A) Blood cell analysis. (B) Liver function indi-cators and blood lipids. (C) Kidney function indicators. Abbrevitions: RBC, red blood cell count; WBC, white blood cell count; PLT, platelet count; HGB, hemoglobin; MCH, mean corpuscular hemoglobin; MCHC, mean corpuscular hemoglobin concentration; HCT, hematocrit; ALB, albumin; ALT, alanine aminotransferase; AST, aspartate aminotransferase; HDL, high-density lipoprotein; TG, triglycerides; TC, total cholesterol; TBIL, total bilirubin; DBIL, direct bilirubin; Scr, serum creatinine; BUN, blood urea nitrogen; UA, uric acid; Cys-C, cystatin C.
- Page 14, line 345: "Hence, a decreased C0/Dose with 344 age was in line with the ontogeny of CYP enzymes involved in SRL metabolism. And a 345 population pharmacokinetics..." This sentence seems to be accidentally divided into two.
Yes. Revisions have been made.
As the comments are more technical, I recommend minor revision.

Reviewer 2 Report
Comments and Suggestions for Authors
The paper entitled: "Therapeutic drug monitoring for sirolimus in children with vascular anomalies: what can we learn from a retrospective study" by Ya-Hui Hu et al. is an interesting retrospective analysis taking into account the TDM of Sirolimus in clinical cases of pediatric vascular disease.
The study is well conducted with shareable patient stratification. However this stratification leads to groups with some having a very reduced number of patients. Such small numbers fall almost into the realm of case reports. We understand that provided the rarity of these diseases, numbers can't be statistically relevant. This has been highlighted in study limitations so we think there is not much to be done to address this in practice.
The paper essentially deals with a series of negative results such as, for instance, the negative correlation between age and BW with C0/Dose ratio at the initial and the last measurements; or the negligible effect of concomitant drugs on C0/Dose ratio. Even the patients’ laboratory results did not change significantly after SRL treatment.
The use of a more structured presentation of results (even a numbered list, for instance) could help provide an at a glance reading of the paper.
Other negative findings were found in C0 between the total response group and the non-response (SD and PD) group.
A high variability in SRL blood concentrations in Chinese children with VAs confirms the consolidated knowledge that SRL is a drug characterized by high variability levels.
Also the conclusion that pediatric patients with older age and a higher BW had a lower C0/Dose ratio is also very predictable based on SLR PK data in the literature.
The finding of inadequacy of the 10-15 ng/mL range for Chinese children and the need to redefine the target reference range for children with VAs are possibly linked also to the analytical technique used for measuring drug concentrations.
It is possible to use state of the art LC-MS/MS techniques for drug concentration assessment.
An analytical chemistry dedicated paragraph in discussion could be useful.
For its negative conclusions the paper is interesting and in perspective useful to inspire future attempts in improving the TDM of SRL in specific patient populations.
Author Response
Reviewer 2: The paper entitled: "Therapeutic drug monitoring for sirolimus in children with vascular anomalies: what can we learn from a retrospective study" by Ya-Hui Hu et al. is an interesting retrospective analysis taking into account the TDM of Sirolimus in clinical cases of pediatric vascular disease.
The study is well conducted with shareable patient stratification. However this stratification leads to groups with some having a very reduced number of patients. Such small numbers fall almost into the realm of case reports. We understand that provided the rarity of these diseases, numbers can't be statistically relevant. This has been highlighted in study limitations so we think there is not much to be done to address this in practice.
The paper essentially deals with a series of negative results such as, for instance, the negative correlation between age and BW with C0/Dose ratio at the initial and the last measurements; or the negligible effect of concomitant drugs on C0/Dose ratio. Even the patients’laboratory results did not change significantly after SRL treatment.
The use of a more structured presentation of results (even a numbered list, for instance) could help provide an at a glance reading of the paper.
Other negative findings were found in C0 between the total response group and the non-response (SD and PD) group.
A high variability in SRL blood concentrations in Chinese children with VAs confirms the consolidated knowledge that SRL is a drug characterized by high variability levels.
Also the conclusion that pediatric patients with older age and a higher BW had a lower C0/Dose ratio is also very predictable based on SLR PK data in the literature.
The finding of inadequacy of the 10-15 ng/mL range for Chinese children and the need to redefine the target reference range for children with VAs are possibly linked also to the analytical technique used for measuring drug concentrations.
It is possible to use state of the art LC-MS/MS techniques for drug concentration assessment.
An analytical chemistry dedicated paragraph in discussion could be useful.
For its negative conclusions the paper is interesting and in perspective useful to inspire future attempts in improving the TDM of SRL in specific patient populations.
We thank you for the positive and constructive feedback that recognizes the contributions of our study.
Indeed, liquid chromatography-tandem mass spectrometry (LC-MS/MS) has been widely applied for the analysis of low molecular weight molecules with the strengths of low interference, good selectivity, high degree of sensitivity, and high throughput. As you concerned, we have established an LC-MS/MS method for SRL and assessed the method consistency between the LC-MS/MS and routine enzyme multiplied immunoassay technique (EMIT) technique for SRL determination in our laboratory [1] . We found that the two methods were closely correlated and that switching between the two methods is feasible. Bland-Altman plots showed a mean concentration overestimation of 4.7 ng/ml and a positive bias of 63.1% generated by the EMIT more than that of by LC-MS/MS.
Reviewing previous studies, immunoassay methods such as microparticle enzyme immunoassay (MEIA), EMIT, chemiluminescent microparticle immunoassay (CMIA), fluorescence polarization immunoassay and chromatography methods like high performance liquid chromatography (HPLC), LC-MS/MS have been used to determine the concentration of SRL. However, few studies evaluated whether the most frequently target therapeutic window 10-15 ng/mL or 5-15 ng/mL in pediatric patients need to be adjusted between different methods for SRL concentration analysis. Hence, through our retrospective study, we endeavor to provide relevant real-world clinical data, thereby encouraging researchers to make more efforts in this field.
Ref:
[1] Zhao YT, Dai HR, Li Y, Zhang YY, Guo HL, Ding XS, Hu YH*, Chen F*. Comparison of LC-MS/MS and EMIT methods for the precise determination of blood sirolimus in children with vascular anomalies [J]. Front Pharmacol. 2022; 13:925018.

Reviewer 3 Report
Comments and Suggestions for Authors
The Manuscript pharmaceuticals-3161973-peer-review-v1
Manuscript pharmaceuticals-3161973-peer-review-v1 describes the therapeutic drug monitoring for sirolimus in Chines children with vascular anomalies.
The manuscript is well-written and within the scope of pharmaceuticals. The manuscript revealed important findings that could be of great value for the optimization of sirolimus doses in children with vascular anomalies.
Major concerns:
1. Authors have used the abbreviation, C0, to represent trough plasma concentrations. C0 is commonly used as an abbreviation for initial plasma concentration at time zero. This could be confusing to the readers. It is recommended to avoid this abbreviation.
2. Figure 1 is duplicated in the manuscript.
3. Patients and methods. Authors need to describe the method of dose calculation. Have doses been calculated based on individual parameters such as body surface area and hepatic functions.
4. Therapeutic drug monitoring is the core of this manuscript as indicated in the title. Therefore, it is crucial to describe how doses were optimized/adjusted based on measured trough concentrations.
5. Results, Characteristics of patients: Authors need to describe body mass index of the patients.
6. The term “dose-corrected C0” is unclear. This term has been used instead of the C0-to-daily dose (C0/Dose) ratio. It is recommended to avoid using the term “dose-corrected C0”.
7. Discussion. Authors need to discuss the impact of differences in body mass index and subsequent effects on the total volume of distribution for dose calculation/optimization.
Author Response
Reviewer 3: The manuscript is well-written and within the scope of pharmaceuticals. The manuscript revealed important findings that could be of great value for the optimization of sirolimus doses in children with vascular anomalies.
Major concerns:
- Authors have used the abbreviation, C0, to represent trough plasma concentrations. C0 is commonly used as an abbreviation for initial plasma concentration at time zero. This could be confusing to the readers. It is recommended to avoid this abbreviation.
Yes. Ctrough has been used to replace C0 in the resubmitted manuscript.
- Figure 1 is duplicated in the manuscript.
Exactly. The duplicated Figure 1 has been removed.
- Patients and methods. Authors need to describe the method of dose calculation. Have doses been calculated based on individual parameters such as body surface area and hepatic functions.
The initial dose of SRL was 0.1 mg/kg/d or 0.8 mg/m2/d, twice daily at 12 h interval, then adjusted according to the target therapeutic window 10-15 ng/mL. We have inserted the corresponding description into the “2.1. Patients” section of the revised manuscript.
- Therapeutic drug monitoring is the core of this manuscript as indicated in the title. Therefore, it is crucial to describe how doses were optimized/adjusted based on measured trough concentrations.
That’s correct. To be honest, clinicians still use empirical therapy for dose adjustment of SRL. Indeed, we are trying to establish a population pharmacokinetic model of sirolimus in children with vascular abnormalities using a nonlinear mixed effects modeling approach. Monte Carlo simulation was employed to propose specific dosage recommendations based on the developed model. This work will be published elsewhere in this or the next year.
- Results, Characteristics of patients: Authors need to describe body mass index of the patients.
Revisions have been made.
- The term “dose-corrected C0” is unclear. This term has been used instead of the C0-to-daily dose (C0/Dose) ratio. It is recommended to avoid using the term “dose-corrected C0”.
That’s correct. Revisions have been made.
- Discussion. Authors need to discuss the impact of differences in body mass index and subsequent effects on the total volume of distribution for dose calculation/optimization.
The detailed description has been shown in the “Discussion” section as follows:
In our study, the median BMI was 16.2 (range 13.0–26.3) kg/m2 and 16.7 (range 12.2–22.4) kg/m2 before and after SRL use, respectively. Some previous studies have reported negative growth impacts associated with SRL in renal transplant patients [30, 31]. Intriguingly, a fairly high proportion (76.7%) of patients with low BMI (< 18.5 kg/m2) were observed before SRL administration, which may be related to the primary disease and prior treatment. In fact, the impact of SRL use on BMI is minimal, no significant differences were found in BMI between before and after SRL treatment (P = 0.9710). This finding was consistent with Wang et al.’ s [32]. Also, the correlation between BMI and Ctrough/Dose ratio was also evaluated during SRL treatment. In the present study, BMI had no relevant effects on the SRL Ctrough/Dose ratio at either initial or last measurements.
Ref:
[30] Rangel, G.A. and G. Ariceta, Growth failure associated with sirolimus: case report. Pediatr Nephrol, 2009. 24(10): p. 2047-50.
[31] Gonzalez, D., et al., Growth of kidney-transplanted pediatric patients treated with sirolimus. Pediatr Nephrol, 2011. 26(6): p. 961-6.
[32] Wang, Y.Y., et al., Long-term safety and influence on growth in patients receiving sirolimus: a pooled analysis. Orphanet J Rare Dis, 2024. 19(1): p. 299.

Round 2
Reviewer 3 Report
Comments and Suggestions for Authors
The revised version of the manuscript entitled "Therapeutic drug monitoring for sirolimus in children with 2 vascular anomalies: what can we learn from a retrospective 3 study" has been improved significantly. It seems that authors have addressed all the comments.